# Effects of Combined Plyometric and Shooting Training on the Biomechanical Characteristics during the Made Jump Shot in Young Male Basketball Players

**DOI:** 10.3390/ijerph20010343

**Published:** 2022-12-26

**Authors:** Marko Radenković, Anja Lazić, Dušan Stanković, Milan Cvetković, Višnja Đorđić, Miloš Petrović, Milena Tomović, Evangelia Kouidi, Adem Preljević, Jovan Marković, Dragana Berić, Marko Stojanović, Miodrag Kocić, Nikola Aksović, Emilija Petković, Milan Čoh, Špela Bogataj, Saša Bubanj

**Affiliations:** 1Faculty of Sport and Physical Education, University of Niš, 18000 Niš, Serbia; 2Faculty of Sport and Physical Education, University of Novi Sad, 21000 Novi Sad, Serbia; 3Research Centre of Movement Sciences, Faculty of Medicine, University of Iceland, 102 Reykjavik, Iceland; 4Sports Medicine Laboratory, Department of Physical Education and Sports Science, Aristotle University of Thessaloniki, 54124 Thessaloniki, Greece; 5Department of Biochemical Science and Sport, State University of Novi Pazar, 36300 Novi Pazar, Serbia; 6Faculty of Pedagogy, University of Kragujevac, 31000 Užice, Serbia; 7Faculty of Sport, University of Ljubljana, 1000 Ljubljana, Slovenia

**Keywords:** explosive strength, jumping performance, biomechanics, plyometric, young players

## Abstract

(1) Background: Shooting performance is one of the most important determinants of basketball success and is strongly influenced by vertical jump performance. A lot of research attention has been paid to training programs that may improve the vertical jump. However, the literature regarding the improvement of accuracy during the jump shot is limited. The aim of this study was to determine the effects of the combination of two training programs on explosive power of the lower extremities during the made jump shot. (2) Methods: A total of 61 male basketball players were assigned into training group (T, *n* = 31, age 15.32 ± 0.65) which was conducting a specific, i.e., experimental training program, and control group (C, *n* = 30, age 16.3 ± 0.71 years) involved in a regular training program. The experimental training program included specific plyometric training with shooting training which lasted for 10 weeks. The obtained data were processed by nonparametric statistics to determine the differences in the vertical jump outcomes, as well as to determine the level of impact of the experimental training program. Wilcoxon and Kruskal–Wallis tests were used. (3) Results: A significant improvement (*p* ≤ 0.05) was noticed in the T group, in every vertical jump variable (flight time, height of the jump, power, and speed of the jump during a jump shot for two and three points), while there was no improvement within the C group. (4) Conclusions: The combination of plyometric and shooting training has a positive impact on the explosive power of the lower extremities during the jump shot.

## 1. Introduction

Basketball is an intermittent, complex sport [1], where shooting performance is one of the most frequent and important technical parts of the game [2]. More precisely, during a 40 min time span, basketball players perform up to 50 jumps [3], with combinations of vertical jumping and shooting elements included in 48.7% of actions [4]. Moreover, when importance of different types of shots were considered, jump shot was found to be the most discriminatory element and the most effective shooting technique, which means that success in basketball and overall performance directly depends on the jump shooting accuracy [5].

It is well known that basketball players have to make a shot under various demanding conditions such as internal and external loads and fatigue [6]. Clearly, it is important to highlight that superior and well-improved strength and conditioning capacities can positively affect shooting performance [5]. More precisely, previous studies have determined that explosive power of lower extremities and elbow extensor isokinetic strength are the main determinants of the long-distance accuracy and that well-developed power and strength may improve the shooting performance [7,8].

Various training approaches were used in order to improve several components of physical attributes in basketball players [9]. However, according to Ziv and Lidor [10], plyometric training is one of the most effective training programs. The plyometric training might be the most effective way to improve physical capacities since it involves movements that are common for real game situations and has high transferability to real game situations [11,12]. Recently it was discovered that plyometric training is effective in improving several variables of basketball performance, such as the height of the vertical jump [13,14] or balance [15,16]. However, to the best of our knowledge, no study has investigated the combination of plyometric and shooting training. Taking into account the importance of well-developed explosive power and complexity of accuracy during the jump shot, it could be of great value to find a way to transfer these abilities into real game situations.

Therefore, the aim of this study was to investigate the effects of a specific training program which included plyometric and shooting drills on explosive power of lower extremities during the made jump shot.

We hypothesized that there is (a) a difference in participants’ results between the initial and final measurement within the same group; (b) an impact of the experimental program on biomechanical variables during the jump shot, and; (c) a difference in participants’ results between the initial and final measurement, and between different groups.

## 2. Materials and Methods

### 2.1. Participants

The sample comprised 61 male basketball players divided into training group (T, *n* = 31, age: 15.3 ± 0.7 years; height: 181.7 ± 5.7 cm; body mass: 70.9 ± 12.3 kg; BMI: 22 ± 3.5) and control group (C, *n* = 30, age: 16.3 ± 0.7 years; height: 187.2 ± 7 cm; body mass: 81.9 ± 10.5 kg; BMI: 23.1 ± 2.5), with basketball experience of more than a year. The T participants were members of the U16 basketball team that competed in the regional “Cadet” league, while the C participants were members of the U17 basketball team that competed in the regional “Triglav” league. Since no differences between T and C subjects were determined in the examined biomechanical characteristics at the initial measurement, there was no need for randomization. All participants signed informed consent forms for involvement in the study. By signing the form, the club, the parents and the participants confirmed that they were familiar with the experimental program. All participants were free of injuries. The study was approved by the Institutional Review Board of the University of Niš, number 8/18-01-006/16-034.

### 2.2. Procedures

The initial assessment was conducted before the beginning of the pre-season period. The experimental program lasted for 10 weeks, and the final assessment was conducted at the end of the program. Prior to the initial and final assessment, the participants applied a standardized warm-up that consisted of 10 min of low-intensity running, 10 min of dynamic stretching, and 5 min of specific basketball movements. Both assessments were performed on hardwood, basketball court, at the same time of the day (9–11 a.m.). Each participant performed a jump shot from the left and right wing and from the central position. Each jump shot was repeated until three successful attempts were obtained, and the participant immediately moved to the next position. The goal was to obtain three made jump shots from each position that were further selected from all attempts (successful and unsuccessful) and analyzed. Hence, the exact number of analyzed made shots was 549 (61 participant × 3 shooting positions × 3 made shots). Shooting positions were set at a 5 m distance for two points and at 6.75 m for three points. Two teammates assisted the shooter by catching and passing him the ball (Figure 1). Experimental program was an addition to the regular training routine. C participants followed the same/regular structure of training program as T participants, but without additional experimental treatment that included a combination of plyometric and shooting exercises. The regular training program was based on tactics, basketball elements, offensive and defensive technique/actions, and dribbling drills implemented by the coaches.

### 2.3. Vertical Jump Assessment

The vertical jump performance was assessed during the jump shooting and tested with an optical measurement system consisting of a transmitting and receiving bar (Optojump, Microgate, Bolzano, Italy). The validity and reliability of Optojump have been confirmed [17]. The extracted outcomes in each trial were the following: flight time of the jump during a jump shot (FLT in s); jump height (JH in cm); power during the jump shot (POW in W/kg); force during the jump shot (FOR in N/kg); and speed during the jump shot (SPE in cm/s).

### 2.4. Experimental Program

The structure of experimental training program consisted of the introductory part of the training, i.e., a warm-up (lasting 10–15 min and including running in a straight line, 4 h 30 m skip forward, 4 h 30 m skip to the side and 4 h 30 m skip backwards), and then static stretching individually or in pairs (for a duration of 4–5 min); the preparatory part of the training (lasting 5 min and aimed to familiarize the participants with the exercises and tasks that will be carried out in the main part of the training); the main part (the participants conducted a training of 50–60 min); and the final part of the training (intended for muscle relaxation and body recovery for 5–10 min). The experimental program included a combination of plyometric and shooting exercises. The plyometric program had five levels of exercise load intensity. The intensity progressed during the weeks—from low intensity in the first week, low/medium during the second, third and fourth, medium/high during the fifth and sixth week, to high intensity training program in the eighth and ninth week and high/medium during the last week. Rest was carried out during the seventh week in order to avoid overtraining (Table 1).

Plyometric training included all kinds of jumps that are similar to the basketball performance (forward bounds, plyo lunges, stance jumps, matrix jumps, depth jumps, backboard touches, toss and catch). Exercises were done bilaterally or unilaterally depending on the week of the experimental program. Additionally, the participants executed exercises for the upper limbs such as various types of push-ups (on the ball, with a basketball as a footrest), dips and various ways of throwing and catching the medicine ball [18].

In contrast to plyometric training, shooting drills were not divided into dosage-based sections (low, low/medium, medium/high, etc.). Instead, shooting program, i.e., drills and tasks were challenging and performed in realistic game conditions (task duration was shortened, distances were increased, number of made shots was increased, passive and active defense was included, etc.). Each training session included four different exercises (e.g., jump shots after one dribbling, jump shot after running in, five in a row, exercise called “seven of seven”, rotating the cones to the basket, jump shot after zig-zag movement, etc.). The experimental training program with exercises and dosing can be found in the work of Radenkovic et al. [19].

### 2.5. Statistical Analysis

Statistical analysis was performed by using the SPSS statistical software (version 20.0, IBM Inc., Armonk, NY, USA) [20]. Due to lack of normality of the data, which is confirmed by the Kolmogorov–Smirnov test, non-parametric procedures were used. For the comparison of the participants’ results between the initial and final measurement within the same group, Wilcoxon test was used. The significance level was set to *p* ≤ 0.05. To determine the impact of the experimental program on biomechanical variables during the jump shot, the statistic *r* (1) that converts z-score into the effect size estimate was used (Field, 2009).
(1)$$r=\frac{Z}{\sqrt{N}}\;.$$

The value “N” represents the number of processed data. In order to obtain the values of “Z”, the difference between the initial and the final measurement was determined using the Wilcoxon test.

For the comparison of the participants’ results between the initial and final measurement and between different groups, Kruskall–Wallis test for independent samples was used. Effect size was estimated by eta squared statistic (η^2^).

## 3. Results

Table 2 shows the differences in T between the initial and final assessment for the variables of the vertical jump during the jump shot for two (2p) and for three (3p) points. Wilcoxon test shows that there are significant differences between measurements in all variables, but with different p levels. Hence, the highest statistical significance can be seen in FLT2p, FLT3p, JH2p, JH3p, POW2p, POW3p, and SPE2p, SPE3p (*p* ≤ 0.00). A slightly smaller, but still statistically significant difference can be seen in FOR3p (*p* ≤ 0.01) and FOR2p (*p* ≤ 0.03). Values from the “Md” column show that there has been an increase in all variables except for FOR2p.

Table 3 shows the level of influence of the experimental program in T for the variables related to explosive power of lower extremities during jump shot for two (2p) and for three (3p) points. The values in column “r” that represents an abbreviation for the non-parametric effect size calculations show that there are different levels of influence (high level—FLT2p, FLT3p, JH2p, JH3p, POW2p, POW3p, SPE2p, SPE3p; low level—FOR2p, FOR3p).

Table 4 shows that the significant difference between the C and T at the initial measurement existed only in SPE3p (*p* ≤ 0.00), while at the final measurement significant differences were determined in all variables (p ranging from *p* ≤ 0.00 to *p* ≤ 0.03) with an exception of FOR2p (*p* ≤ 0.84).

## 4. Discussion

The aim of this study was to determine the influence of the combination of two types of training on the explosive power of lower extremities during the made jump shot in young basketball players. The results obtained by T in our research revealed significant difference between the initial and the final measurement in all analyzed variables. This means that there was an impact of the experimental program. In addition, it can be noticed that the results from Table 2 and Table 3 correspond, as the level of influence and level of statistical significance are lower for the same variables (FOR2p and FOR3p). Results obtained by C in our study are fairly expected, and in accordance with the study of de Villarreal. et al. (2021) [21]. Namely, no improvements were also reported in height of the vertical jump when C was considered [21]. The reason for such a statement lies in the fact that the C participants were involved with a training protocol without additional experimental treatment in comparison to the T peers. The emphasis of C plan and program was placed not on the impact of experimental program on biomechanical variables, but on practicing tactics, basketball elements, technique/actions in attack/defense and dribbling.

Basketball is a relatively complex sport that implies well-developed physical capacity, especially aerobic, anaerobic capacity, upper-body and lower-body power, agility, and change of direction speed [4,22,23], as well as well-developed technical elements such as shooting accuracy [2]. However, implementing separate training protocols for improving the physical capacity may be time consuming [24]. In accordance with this finding, the combination of two types of training that we implemented may be beneficial in order to simultaneously improve two important factors (vertical jump performance and shooting accuracy) for successful playing. In favor of our findings, some authors [25,26] suggested that more complex training programs may be more beneficial for improving vertical jump performance than plyometric training alone, taking into account the complex nature of vertical jumps present in basketball game [27]. Additionally, Knudson [28] highlighted the importance of correlation between the explosive power of the lower extremities and shooting accuracy, considering the fact that more accurate shooters let out the ball at a greater height. Even though our experimental training program is not a classic plyometric training (combined with a shooting training), it has been found that, besides improving the height of the vertical jump, it has also had a positive effect on other explosive power variables: flight time of the jump, power, force, and speed of the jump.

Shooting accuracy is one of the most important determinants of overall performance in basketball [29]. Results of our study indicate that at the end of the experimental program, there were improvements in overall accuracy. However, due to the poor reliability of the tests for accessing the shooting accuracy and complexity of this ability, we cannot claim that better performance was the consequence of the shooting training. On the contrary, better shooting performance is probably the result of improved vertical jump variables. More precisely, vertical jump performance strongly correlates with dynamic shooting tests, especially from longer distances [6]. In general, these findings are not surprising due to the biomechanical and physiological aspects of jump shot [30]. Jump shot is the dynamic action and it is usually performed in high-intensity conditions where the importance of producing force rapidly is the crucial factor, while the physiological basis similar to the vertical jump is logical and obvious. In more detail, basketball players who present better vertical jumping abilities perform the jump shot with a lower release velocity, which provides them with more time for executing the correct shooting technique [28]. Our results are consistent with this finding. On the other hand, the players with inferior jumping performance are not able to generate the adequate force intensity as fast as it is possible [28]. Instead, they promote compensatory mechanisms and movements that may negatively affect shooting accuracy [31] in order to improve segmental velocity and to achieve the greater ball distance [28].

We acknowledge that our study has some limitations. Firstly, our study is not generalizable beyond the study sample due to the size of the sample, as well as the level of the competition. Secondly, while the improved vertical jump performance is a consequence of the plyometric program, we cannot claim that better shooting performance is also the consequence of the plyometric, shooting training or the combination of the two types of training programs. Nevertheless, we consider our results as promising, and strongly encourage sport scholars to further investigate the role of different types of training programs in order to improve overall basketball performance.

## 5. Conclusions

In conclusion, this study provides evidence that combined plyometric and shooting training is an effective method for improving flight time, power, force and speed of the jump during a jump shot in young male basketball players. The shooting performance requires superior levels of the explosive power outputs. The importance of biomechanical variables plays a major role in the jump shot performance; the whole musculoskeletal system is involved. In addition, there are many factors that affect the accuracy, but one of the most important of them is greater height during vertical jump. Further studies are needed in order to determine the ways to improve accuracy during the vertical jump due to the importance of this element in the real game situations.

## Figures and Tables

**Figure 1 ijerph-20-00343-f001:**
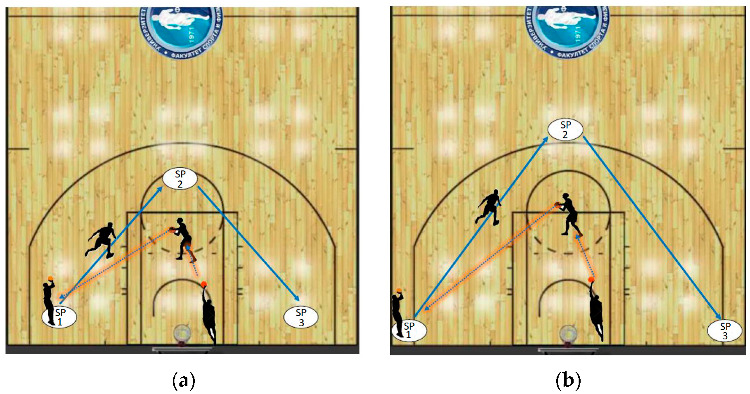
Jump shot assessment, two-points (**a**) and three points (**b**) shooting positions (SP).

**Table 1 ijerph-20-00343-t001:** Plyometric program dosing guidelines.

Week	Load Intensity	Number of Exercises	Sets	Repetitions per Training Session
1	Low	5	1	4–8
2–4	Low/medium	5–6	1–2	4–8
5–6	Medium/high	5–6	1–2	5–10
7	Rest	/		
8–9	High	6	3	5–10
10	High/medium	5	2–3	5–10

**Table 2 ijerph-20-00343-t002:** Differences in experimental (T) and in control group (C) between the initial and final measurement calculated by the Wilcoxon test.

	T					
	Z	Sig.	Md_I	IQR_I	Md_F	IQR_F
FLT2p-FLT2pF (s)	−9.67	<0.001	0.27	0.21–0.32	0.30	0.28–0.37
FT3p–FT3pF (s)	−10.88	<0.001	0.34	0.28–0.37	0.38	0.34–0.41
JH2p–JH2pF (cm)	−13.53	<0.001	9.60	5.5–12.6	15.60	12–17.8
JH3p–JH3pF (cm)	−14.14	<0.001	14.10	9.7–16.9	21.00	17.8–24.3
POW2p–POW2pF (W/kg)	−10.46	<0.001	6.99	5.22–8.22	8.54	7.42–9.51
POW3p–POW3pF (W/kg)	−12.96	<0.001	8.71	7.08–9.74	10.58	9.72–11.34
FOR2p–FOR2pF (N/kg)	−2.20	0.028	0.45	0.32–0.53	0.44	0.38–0.5
FOR3p–FOR3pF (N/kg)	−2.66	0.008	0.58	0.49–0.67	0.60	0.54–0.66
SPE2p–SPE2pF (cm/s)	−13.76	<0.001	16.00	14.7–17.37	19.58	18.16–20.84
SPE3p–SPE3pF (cm/s)	−13.69	<0.001	14.69	13.64–15.84	17.84	16.69–18.94
	**C**					
	**Z**	**Sig.**	**Md_I**	**IQR_I**	**Md_F**	**IQR_F**
FLT2p–FLT2pF (s)	−0.823	0.410	0.28	0.22–0.33	0.28	0.22–0.33
FT3p–FT3pF (s)	−1.273	0.203	0.33	0.28–0.38	0.34	0.28–0.39
JH2p–JH2pF (cm)	−0.928	0.354	9.65	5.65–13.3	9.70	5.48–13.43
JH3p–JH3pF (cm)	−1.537	0.124	13.40	9.68–18.1	14.25	9.75–18.08
POW2p–POW2pF (W/kg)	−0.396	0.692	7.26	5.51–8.56	7.20	5.56–8.65
POW3p–POW3pF (W/kg)	−1.426	0.154	8.52	7.13–10.17	8.90	7.37–10.09
FOR2p–FOR2pF (N/kg)	−0.19	0.849	0.43	0.32–0.53	0.45	0.32–0.53
FOR3p–FOR3pF (N/kg)	−0.799	0.425	0.56	0.46–0.69	0.58	0.49–0.67
SPE2p–SPE2pF (cm/s)	−0.221	0.825	16.00	15.05–17.53	16.00	14.83–17.76
SPE3p–SPE3pF (cm/s)	−0.52	0.603	15.240	13.55–16.76	14.780	13.72–16.25

2p—jump shot for two points; 3p—jump shot for three points; I—initial measurement; F—final measurement; Sig.—level of *p* value; Md—Median, IQR—inter-quartile range.

**Table 3 ijerph-20-00343-t003:** Influence of specific training program in the experimental group (T).

	N	√N	Z	r
FLT2p (s)	549	23.62	9.67	0.41
FLT3p (s)	549	23.62	10.88	0.46
JH2p (cm)	549	23.62	13.53	0.57
JH3p (cm)	549	23.62	14.14	0.60
POW2p (W/kg)	549	23.62	10.46	0.44
POW3p (W/kg)	549	23.62	12.96	0.55
FOR2p (N/kg)	549	23.62	2.20	0.09
FOR3p (N/kg)	549	23.62	2.66	0.11
SPE2p (cm/s)	549	23.62	13.76	0.58
SPE3p (cm/s)	549	23.62	13.69	0.58

2p—jump shot for two points; 3p—jump shot for three points; N—the number of processed data; √N—square root of N; Z—values from Table 1; r—the level of influence.

**Table 4 ijerph-20-00343-t004:** Differences between control (C) and experimental (T) group at the initial and final measurement calculated by the Kruskall–Wallis test for independent samples.

	Initial				Final			
	Chi-Squ.	df	Sig.	η^2^	Chi-Squ.	df	Sig.	η^2^
FLT2p (s)	1.135	1	0.287	0.002	44.799	1	<0.001	0.082
FLT3p (s)	0.316	1	0.574	0.001	63.302	1	<0.001	0.116
JH2p (cm)	0.564	1	0.453	0.001	140.308	1	<0.001	0.256
JH3p (cm)	0.044	1	0.833	0.000	195.419	1	<0.001	0.357
POW2p (W/kg)	1.729	1	0.189	0.003	59.307	1	<0.001	0.108
POW3p (W/kg)	0.587	1	0.443	0.001	131.295	1	<0.001	0.240
FOR2p (N/kg)	0.163	1	0.686	0.000	0.097	1	0.756	0.000
FOR3p (N/kg)	0.790	1	0.374	0.001	4.569	1	0.033	0.008
SPE2p (cm/s)	2.572	1	0.109	0.005	196.610	1	<0.001	0.359
SPE3p (cm/s)	7.094	1	0.008	0.013	165.613	1	<0.001	0.302

2p—jump shot for two points; 3p—jump shot for three points; Chi-Squ.—Chi-Square; df—degrees of freedom; Sig.—the level of significance, η^2^—effect size.

## Data Availability

Not applicable.

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
