# Peer review of "Effects of Combined Plyometric and Shooting Training on the Biomechanical Characteristics during the Made Jump Shot in Young Male Basketball Players"

_ijerph, 2022, doi:10.3390/ijerph20010343_

Round 1
Reviewer 1 Report
1) In the abstract, please mention which statistical tests were used.
2) In the introduction, we need more references for the effects of plyometric training in vertical jump and especially for basketball skills.
3) Is any reference for order of load intensity of polymetric training in Table 1?
4) As well, formula in Figure 1 needs reference …. Please clarify it.
5) In Results, tables should be placed after its results. For example, Table 3 should be shifted to after first paragraph of the results. As well, for other tables in the results section.
6) References could be improved buy adding more recent and relevant studies.
7) Conclusion should be based on your findings and practical implications.
Good luck.
Author Response
Dear,
We are pleased to submit a revised version of our paper titled "Effects of Combined Plyometric and Shooting Training on the Biomechanical Characteristics During the Made Jump Shot in Young Basketball Players".
We would like to thank you for taking time to review the manuscript and providing constructive criticism. We did our best to address all comments and we feel that the quality of the paper has improved.
Please, find out point by point responses to comments below. The changes in the manuscript are visible as track changes, according to the instructions of the journal.
____________________________________________________________________________
Dear Editor,
We are pleased to submit a revised version of our paper.
Manuscript ID: International Journal of Environmental Research and Public Health; ijerph-2016719
Title: Effects of Combined Plyometric and Shooting Training on the Biomechanical Characteristics During the Made Jump Shot in Young Basketball Players
The authors would like to thank the reviewers for taking their time to review the manuscript and providing constructive criticism. We did our best to address all comments and we feel that the quality of the paper has improved.
Please, find out point by point responses to comments below. The changes in the manuscript are visible as track changes, according to the instructions of the journal.
Reviewer: In the abstract, please mention which statistical tests were used.
Our response: Thank you for your comment. Statistical analyses were added in the abstract section.
Reviewer: In the introduction, we need more references for the effects of plyometric training in vertical jump and especially for basketball skills.
Our response: Thank you for the suggestion. We agree with this suggestion, and we have added more, relevant references.
Reviewer: Is any reference for order of load intensity of polymetric training in Table 1?
Our response: Yes, thank you for this comment. We have added reference for load intensity of plyometric training.
Reviewer: As well, formula in Figure 1 needs reference …. Please clarify it.
Our response: Thank you for the suggestion. We have clarified the reference related to Figure 1, while the formula itself is not presented as figure anymore, but as an equitation.
Reviewer: In Results, tables should be placed after its results. For example, Table 3 should be shifted to after first paragraph of the results. As well, for other tables in the results section.
Our response: Thank you for your suggestion. We took your advice and re-wrote the results section and placed tables below text.
Reviewer: References could be improved buy adding more recent and relevant studies.
Our response: Thank you for this useful point. We have added some recent and relevant studies.
Reviewer: Conclusion should be based on your findings and practical implications.
Оur response: Thank you for the useful suggestion. We tried to write conclusion based on our findings.
Reviewer 2 Report
The aim of this study was to determine the effects of the combination of two training programs on explosive power of the lower extremities during the jump shot in adolescent players. There are issues with the statistical approach utilised and the entire manuscript needs to be revised for grammar and missing words. Please see my specific comments below.
Title: Include gender of population studied.
Abstract
L36-38: Word missing.
L38-39: Include gender.
Introduction
Needs to be revised for grammar.
Materials and Methods
Gender of the participants should be included.
L83-85: I think using ES and CS is confusing. ES is typically used as an abbreviation for effect size. Consider using alternative abbreviations like T for training group and C for control. Additionally, why not use the term “group” rather than “subsample”?
L83-85: Age is included but mass and height should also be included.
There is no information about how participants were allocated into groups.
Was the control group older than the training group?
The study design should also be clearly stated.
What did the control group do?
L102: Typo.
The jump shot assessment needs to be clearer; can a figure be included to correspond with the text?
L110-112: These abbreviations are very hard to follow; I don’t think they are actually necessary. Use the full word.
L110-111: Describe the equations use to estimate power and force. What does speed represent in this context?
Table 1 – The numerical values are too messy. Use a different column for number of exercises.
L128: More detail required here. How many exercises and what is meant by competitions?
L136: Which version of SPSS?
How was normality assessed?
Was any effect size calculated for the Kruskal-Wallis test?
Results
Table 2 – This table needs to be changed. The text should precede the table appearing. Use full words rather than abbreviations, 2P and 3P can be retained. The p-values should be reported to three decimal places. The p-value is never 0.00. Present the median alongside their interquartile range and use separate columns for pre and post.
Table 3 – How is N=558?
A Wilcoxon test should be performed on the control group data as well.
You should test the differences in the experimental group versus the differences in the control group.
Discussion
L180-181: This is not correct, only by comparing the difference to the difference in the control group can you establish cause and effect.
Author Response
Dear,
We are pleased to submit a revised version of our paper titled "Effects of Combined Plyometric and Shooting Training on the Biomechanical Characteristics During the Made Jump Shot in Young Basketball Players".
We would like to thank you for taking time to review the manuscript and providing constructive criticism. We did our best to address all comments and we feel that the quality of the paper has improved.
Please, find out point by point responses to comments below. The changes in the manuscript are visible as track changes, according to the instructions of the journal.
_______________________________________________________________________________
Dear Editor,
We are pleased to submit a revised version of our paper.
Manuscript ID: International Journal of Environmental Research and Public Health; ijerph-2016719
Title: Effects of Combined Plyometric and Shooting Training on the Biomechanical Characteristics During the Made Jump Shot in Young Basketball Players
The authors would like to thank the reviewers for taking their time to review the manuscript and providing constructive criticism. We did our best to address all comments and we feel that the quality of the paper has improved.
Please, find out point by point responses to comments below. The changes in the manuscript are visible as track changes, according to the instructions of the journal.
The aim of this study was to determine the effects of the combination of two training programs on explosive power of the lower extremities during the jump shot in adolescent players. There are issues with the statistical approach utilised and the entire manuscript needs to be revised for grammar and missing words. Please see my specific comments below:
Reviewer: Title: Include gender of population studied.
Our response: Thank you for your comment. We have included gender, both in the title, and in the section Materials and Methods.
Reviewer: Abstract: L36-38: Word missing; L38-39: Include gender.
Our response: Thank you for the comment. We have added missing word “of” and have included gender within the abstract section.
Reviewer: Introduction. Needs to be revised for grammar.
Our response: Thank you for the suggestion. We have revised grammar for the whole manuscript.
Reviewer: Materials and Methods. Gender of the participants should be included. L83-85: I think using ES and CS is confusing. ES is typically used as an abbreviation for effect size. Consider using alternative abbreviations like T for training group and C for control. Additionally, why not use the term “group” rather than “subsample”?
Our response: Thank you for this suggestion. Gender of the participants have been included. We modified term “subsample” to term “group”, and used different abbreviation for groups, as you indicated.
Reviewer: L83-85: Age is included but mass and height should also be included.
Our response: Thank you for the suggestion. We have added body mass and height parameters in the Participants section.
Reviewer: There is no information about how participants were allocated into groups?
Our response: Thank you for this remark. We have added an explanation within the section Participants: the sample is divided in two groups based on club affiliation and leagues in which they competed.
Reviewer: Was the control group older than the training group?
Our response: Yes, control group was older than the training group, 16.3 ± 0.7 vs 15.3 ± 0.7 years, respectively.
Reviewer: The study design should also be clearly stated.
Our response: Thank you for the suggestion. We gave in depth and clarified study design within the section Procedures.
Reviewer: What did the control group do?
Our response: We appreciate you bringing this to our attention. We have added the part about involvement of the control group: Control group followed a regular training program based on tactics, offensive and defensive technique, and fitness/strength drills designed by their coaches.
Reviewer: L102: Typo.
Our response: Thank you for recognizing this typing mistake. We have re-written the word “unsuccessful”.
Reviewer: The jump shot assessment needs to be clearer; can a figure be included to correspond with the text?
Our response: Thank you for the suggestion. We inserted figure 1 related to jump shot assessment, and two-points (a) and three points (b) shooting positions (SP).
Reviewer: L110-112: These abbreviations are very hard to follow; I don’t think they are actually necessary. Use the full word. L110-111.
Our response: Thank you for this remark. We have renamed the variable abbrevations to make them easier to follow in the text (we excluded prefix “OJ”). We used the full name of the variable at their first mention in the text, while after that we used the abbreviations.
Reviewer: Describe the equations use to estimate power and force. What does speed represent in this context?
Our response: Power can be calculated from the equitation Power = Work / time. The mechanical work performed to accomplish a vertical jump can be determined by using the jump height distance that is measured (using Work = Force x Distance where Force = Mass x Acceleration). Beyond the Newtonian mechanics, the speed indicates the contractile properties of the extensor muscles during the vertical jump, that is, the level of explosive power of the basketball player.
Reviewer: Table 1 – The numerical values are too messy. Use a different column for number of exercises.
Our response: Thank you for this remark. We have added two additional columns for sets and number of repetitions.
Reviewer: L128: More detail required here. How many exercises and what is meant by competitions?
Our response: In contrast to plyometric training, shooting drills were not divided into dosage-based sections (low, low/medium, medium/high, etc.). Instead, shooting program, i.e., drills and tasks were challenging and performed in realistic game conditions (task duration was shortened, distances were increased, number of made shots was increased, passive and active defense was included, etc.). Each training session included four different exercises (e.g. jump shots after one dribbling, jump shot after running in, five in a row, exercise called ‘seven of seven’, rotating the cones to the basket, jump shot after zig-zag movement, etc.). The training program with exercises and dosing can be found in the work of Radenkovic et al. [19].
Reviewer: L136: Which version of SPSS?
Our response: Thank you for the observation. We added SPSS version 20, and reference as well.
Reviewer: How was normality assessed?
Our response: Due to lack of normality of the data, which is confirmed by Kolmogorov-Smirnov test, non-parametric procedures were used. We added missing explanation.
Reviewer: Was any effect size calculated for the Kruskal-Wallis test?
Our response: Effect size was estimated by eta squared statistic (η2). Values are added within table 4.
Reviewer: Results. Table 2 – This table needs to be changed. The text should precede the table appearing. Use full words rather than abbreviations, 2P and 3P can be retained. The p-values should be reported to three decimal places. The p-value is never 0.00. Present the median alongside their interquartile range and use separate columns for pre and post.
Our response: Thank you for pointing this out. We have modified Table 2 according to your suggestions.
Reviewer: Table 3 – How is N=558?
Our response: N is the number of succesfull attempts, every successful shot was considered as sample element, and not a player.
Reviewer: A Wilcoxon test should be performed on the control group data as well.
Our response: Thank you for pointing this out, we extended Table 2 with results related to control group.
Reviewer: You should test the differences in the experimental group versus the differences in the control group.
Our response: According to the results of Kruskal Wallis test performed on the initial measurements, it was obvious that two groups are balanced regarding the tested variables, so we concluded that possible differences in final measurements will be sufficient proof of significant impact of experimental program.
Reviewer: Discussion. L180-181: This is not correct, only by comparing the difference to the difference in the control group can you establish cause and effect.
Our response: L180-181 According to the results of Kruskal Wallis test performed on the initial measurements, it was obvious that two groups are balanced regarding the tested variables, so we concluded that possible differences in final measurements will be sufficient proof of significant impact of experimental program.
Reviewer 3 Report
The aim of this study was to investigate the effects of combined plyometric and shooting training on explosive power of lower extremities during the jump shot in young basketball players. In the introduction, the authors indicate that the effect of plyometric training on improving vertical jump height is scientifically documented. Simultaneously, they justify undertaking their research by pointing out that the effect of combined plyometric and shooting training on lower limb explosive power has not previously been studied. Unfortunately, such a designed study does not allow answering the question of whether the improvement in jump performance is a result of combined plyometric and shooting training, or solely plyometric training. The latter option seems more likely, and this will deprive this work of originality.
The description of the methodology is unclear. The authors write: “The goal set in front of the participant was to obtain three made jump shots from each position, that were further selected from all successful and unsuccessful attempts and analyzed”. Were all jumps analyzed or only selected ones? The easiest way would be to indicate the exact number of the analyzed jumps.
There is no explanation for the decision to stop training in the seventh week. Did the players from the control group also have a break?
There is also no description of the training of the control group. Did the players from this group not train lower limb motor skills or shooting for 10 weeks?
The description of the statistical analysis is vague and incomplete. Why was it decided to use non-parametric tests? Was the distribution of the data checked for normality? What does the parameter expressed by the letter "r" mean? In statistics, usually the symbol "r" stands for Pearson's correlation coefficient, but in this work it is not explained what "r" means. In addition, the use of this parameter seems unjustified, because what it was supposed to test, according to the authors, was verified with the Wilcoxon test. The authors also did not indicate at what level they considered statistical significance.
The presentation of the results is not very clear. The values of the variables in the pre-test and follow-up are missing. The authors only presented the differences and their significance.
Author Response
Dear,
We are pleased to submit a revised version of our paper titled "Effects of Combined Plyometric and Shooting Training on the Biomechanical Characteristics During the Made Jump Shot in Young Basketball Players".
We would like to thank you for taking time to review the manuscript and providing constructive criticism. We did our best to address all comments and we feel that the quality of the paper has improved.
Please, find out point by point responses to comments below. The changes in the manuscript are visible as track changes, according to the instructions of the journal.
______________________________________________________________________________
Dear Editor,
We are pleased to submit a revised version of our paper.
Manuscript ID: International Journal of Environmental Research and Public Health; ijerph-2016719
Title: Effects of Combined Plyometric and Shooting Training on the Biomechanical Characteristics During the Made Jump Shot in Young Basketball Players
The authors would like to thank the reviewers for taking their time to review the manuscript and providing constructive criticism. We did our best to address all comments and we feel that the quality of the paper has improved.
Please, find out point by point responses to comments below. The changes in the manuscript are visible as track changes, according to the instructions of the journal.
Reviewer: The aim of this study was to investigate the effects of combined plyometric and shooting training on explosive power of lower extremities during the jump shot in young basketball players. In the introduction, the authors indicate that the effect of plyometric training on improving vertical jump height is scientifically documented. Simultaneously, they justify undertaking their research by pointing out that the effect of combined plyometric and shooting training on lower limb explosive power has not previously been studied. Unfortunately, such a designed study does not allow answering the question of whether the improvement in jump performance is a result of combined plyometric and shooting training, or solely plyometric training. The latter option seems more likely, and this will deprive this work of originality.
Our response: Thank you for your constructive comment. Yes, we cannot claim that improvements in jump performance is neither due to plyometric training or shooting training. Moreover, we cannot claim that better shooting performance is also the consequence of the plyometric, shooting training or the combination of the two types of training programs, and that is the limitation of our study, as stated within the Discussion section.
Reviewer: The description of the methodology is unclear. The authors write: “The goal set in front of the participant was to obtain three made jump shots from each position, that were further selected from all successful and unsuccessful attempts and analyzed”. Were all jumps analyzed or only selected ones? The easiest way would be to indicate the exact number of the analyzed jumps.
Our response: Only made jumps shots are selected and considered for analysis. We added within the section Procedures exact number (549=61 participant x 3 shooting positions x 3 made shots).
Reviewer: There is no explanation for the decision to stop training in the seventh week. Did the players from the control group also have a break?
Our response: Thank you for the comment. Rest was carried out during the seventh week, in order to avoid overtrainig.
Reviewer: There is also no description of the training of the control group. Did the players from this group not train lower limb motor skills or shooting for 10 weeks?
Our response: Thank you for this observation. We missed out writing this part. Control group followed a regular training program based on tactics, offensive and defensive technique, and fitness/strength drills designed by their coaches, and had no break during the seventh week of their program.
Reviewer: The description of the statistical analysis is vague and incomplete. Why was it decided to use non-parametric tests? Was the distribution of the data checked for normality? What does the parameter expressed by the letter "r" mean? In statistics, usually the symbol "r" stands for Pearson's correlation coefficient, but in this work it is not explained what "r" means. In addition, the use of this parameter seems unjustified, because what it was supposed to test, according to the authors, was verified with the Wilcoxon test. The authors also did not indicate at what level they considered statistical significance.
Our response: We agree with your remark. We added description considering normality of distribution. Effect sizes (r) are calculated due to more detailed (in better resolution) description of the differences.
Reviewer: The presentation of the results is not very clear. The values of the variables in the pre-test and follow-up are missing. The authors only presented the differences and their significance.
Our response: We agree with your remark. The measures of central tendency (median) and measures of dispersion (inter-quartile range) for both groups and for their initial and final measurements are added into the table 2.